# Genome-Wide Association Study for Non-Photochemical Quenching Traits in *Oryza sativa* L.

**Youbo Wei** [1,†]**, Sicheng Liu** [1,†]**, Dongliang Xiong** [1]**, Zhuang Xiong** [1]**, Zuolin Zhang** [2]**, Fei Wang** [1] **and Jianliang Huang** [1,*]

[1] National Key Laboratory of Crop Genetic Improvement, Ministry of Agriculture Key Laboratory of Crop Ecophysiology and Farming System in the Middle Reaches of the Yangtze River, College of Plant Science and Technology, Huazhong Agricultural University, Wuhan 430070, China
[2] Institute of Food Crops, Hubei Academy of Agricultural Sciences, Wuhan 430070, China
[*] Correspondence: jhuang@mail.hzau.edu.cn; Tel.: +86-2787284131
[†] These authors contributed equally to this work and should be considered co-first authors.

**Abstract:** Manipulating the photoprotective mechanism has been demonstrated to be an effective way to enhance the photosynthetic productivity of crop plants. NPQ(T) is a chlorophyll fluorescence parameter for rapid estimation and imaging of non-photochemical quenching (NPQ) of excitons in the photoprotective mechanism. However, the variation and genetic basis of NPQ(T) are rarely reported in the *Oryza sativa* L. In this study, we collected 173 rice cultivars and investigated the NPQ(T) value. We found that the NPQ(T) has a wide variation, although it had not been under-selected in the different subspecies. A genome-wide association study (GWAS) utilizing 1,566,981 high-quality SNPs identified three significant associated signals on chromosomes 02, 05, and 07. Furthermore, one likely candidate gene *Os02g0184100*, underlying the associated signal on chromosome 02, was uncovered by identifying the expression pattern in flag leaves and testing the correlation between functional polymorphisms and phenotypic variation. The significant SNPs and candidate genes identified in this study provide us a comprehensive understanding of the genetic architecture of NPQ(T) and could be used for genetic improvement of rice photoprotection.

**Keywords:** rice; chlorophyll fluorescence; NPQ/NPQ(T); genome-wide association study; photosynthesis

## 1. Introduction

Rice (*Oryza sativa* L.) is the staple food for more than half of the world's population. With the increase in the global population and the decrease in arable land, increasing grain yield is still the primary goal of rice breeders today [1]. The grain yield of rice is mainly determined by biomass accumulation and harvest index. During the green revolution in the early 1960s, the yield of rice increased spectacularly by the introduction of dwarfing traits into the plants, which improved the lodging resistance and harvest index [2–5]. In the past decades, the yield has been further increased by ideotype and heterosis breeding.

Besides the above approach, coordinated regulation of photosynthesis, converting light energy to chemical energy, is another important way to increase yield and tolerance to environmental stress in rice [6]. There are many factors that affect the photosynthesis of crops, such as the morphology and structure of the leaves, the absorption and transmission of light energy, and the photosynthetic carbon cycle [7]. Chlorophyll content and parameters derived from gas exchange measurements and chlorophyll fluorescence have been widely used to reflect photosynthetic capacity [8–10]. Plants have evolved regulatory non-photochemical quenching (NPQ) mechanisms to eliminate damage from excessive light by dissipating the excessive excitation energy as heat in PSII antenna complex [11–13]. Moreover, NPQ has been one of the common parameters of chlorophyll fluorescence and is currently the most advanced non-invasive measurement of plant photosynthetic energy use efficiency [14]. As the measurement of NPQ is complicated, NPQ(T) is a new chlorophyll

fluorescence parameter for the rapid estimation and imaging of NPQ, and can be obtained without dark adaptation in a shorter measurement time [15]. This promotes NPQ(T) as more convenient for field measurement.

With the feasibility of genotyping numerous lines using high-throughput sequencing, linkage mapping and genome-wide association studies (GWAS) have been a powerful approach to identify natural variation underlying complex traits in crops, mainly including morphological characteristics, yield, and grain quality [16–18]. Moreover, several studies had been performed to identify the genetic basis of NPQ by linkage mapping and GWAS in plants, such as *Arabidopsis* [19,20], soybean [21], and rice [22,23]. Additionally, a proposed mechanism underlying the transcriptional regulation of NPQ was established in rice [24]. The associated loci and genes may help enhance photoprotection and improve photosynthesis in plants. However, few loci or genes are reported to be involved in the natural variation of NPQ(T) in rice.

In the present study, we performed GWAS for NPQ(T) using a diverse collection of 173 rice varieties, based on 112,408,956 single-nucleotide polymorphisms (SNPs) derived from genome resequencing. We further identified one candidate gene and possible causal polymorphisms for NPQ(T) by integrating gene annotation, expression pattern, and haplotype analysis. Our findings provide some insight into the genetic basis of NPQ(T), which would be useful in improving photosynthesis efficiency in rice.

## 2. Materials and Methods

### 2.1. Plant Materials

The association panel consisted of a diverse collection of 173 *Oryza sativa* accessions, which includes 78 accessions from the Mini Core Collection of Huazhong Agricultural University and 95 accessions from the 3000 Rice Genomes Project. The details of the accession, including accession name, country of origin, and subpopulation identity, are shown in Supplementary Table S1.

### 2.2. Experimental Design and Measurement of NPQ(T)

About 200 g seeds was sown in the field of Huazhong Agricultural University, Wuhan, China, on May 18th of 2019. One-month-old seedlings were transplanted into 1 m × 2 m plots, with one plant per hill at 0.20 × 0.25 m spacing. Fertilizers applied to all plots were 180 kg N ha$^{-1}$, 60 kg P$_2$O$_5$ ha$^{-1}$, and 120 kg K$_2$O ha$^{-1}$. A local plot trial management was performed, which includes irrigation, fertilization, and disease and pest control. Weather data (daily maximum/minimum temperature, rainfall, relative humidity, and sunshine hours) for the whole growing season are shown in Supplementary Figure S1. Five plants of each accession in the middle of the plot were selected to investigate the NPQ(T) values during the heading stage. In order to better reflect the average NPQ(T) level of rice flag leaf and make the leaf cover the whole sensor to obtain more reliable data, NPQ(T) was measured in the middle part (1/3~2/3) of the flag leaf between 8:30 and 11:30 a.m. on a sunny day, using a portable chlorophyll fluorometer (MultispeQ v1.0) [25], with agreement of no open/close (https://www.photosynq.org/protocols/leaf-photosynthesis-multispeq-v1-0-no-open-close (accessed on 10 August 2022)).

### 2.3. DNA Extraction, Sequencing and Data Processing

DNA was extracted from fresh leaves of field-grown plants using a modified CTAB method [26]. Whole-genome DNA sequencing was performed on the Illumina HiSeq-2000 platform by Personalbio (Shanghai, China). Paired-end 150 bp reads were mapped to the rice cultivar Nipponbare reference genome (https://www.ebi.ac.uk/ena/data/view/GCA_001433935.1 (accessed on 10 August 2022)) using BWA (V0.7.8) with the default parameters. After alignment, genomic variants (in GVCF format for each accession) were identified using the Genome Analysis Toolkit (GATK) software [27], with the Haplotype Caller module and GVCF model. The raw variants were further filtered with the following parameters: depth for each individual ≥5, genotype quality for each individual ≥5, minor

allele frequency (MAF) $\geq$ 0.01, and miss rate $\leq$0.2. The identified SNPs were further annotated using the ANNOVAR software (version 20 May 2013) [28].

### 2.4. Population Genetics Analysis

All 1,566,981 identified SNPs were used to build a phylogenetic tree and perform principal coordinate analysis (PCA). The individual-based neighbor-joining (NJ) tree was constructed using the TreeBest software (v1.9.2), based on the *p*-distance and with 1000 bootstrap replicates [29]. PCA was conducted using the GCTA software [30]. First, the genetic relationship matrix (GRM) was obtained using the parameter '–make-grm'. Then, the top two principal components were estimated using the parameter '–pca3'. To estimate LD in our rice population, the squared correlation coefficient ($r^2$) between pairwise SNPs was computed using PopLD decay [31], with parameters in the program set as '-MaxDist 1000 kb-MAF 0.05-Miss 0.1'. The $r^2$ value was calculated for pairwise markers in a 1000 kb window and averaged across the whole genome. Haplotype blocks were detected using the plink software package [32], with the following parameters: '–blocks no-pheno-req– blocks-max-kb 1000–blocks-min-maf 0.05–blocks-strong-lowci 0.70–blocks-strong-highci 0.98–blocks-recomb-highci 0.90–blocks-inform-frac 0.95'.

### 2.5. Genome-Wide Association Analyses

Genome-wide association study (GWAS) was performed using a mixed linear model (MLM) in the EMMAX (beta version) package [33]. The matrix of pairwise genetic distances, calculated by EMMAX, was used as the variance–covariance matrix of random effects. Significant *p*-value thresholds ($p = 10^{-6}$) were set to control the genome-wide type 1 error rate, which was calculated by 1/n (total SNPs). PVE of 100 kb was filtered out before and after the peak signal. The Manhattan and quantile–quantile (Q-Q) plots of GWAS results were generated in R software.

### 2.6. Analysis of Candidate Genes

To identify the putative candidate gene underlying the associated peak on chromosome 02, a candidate region surrounding the peak signal was selected using an $r^2 > 0.8$. The rice gene annotation and expression profile (https://ricexpro.dna.affrc.go.jp/ (accessed on 10 August 2022)) were used to uncover the candidate genes. The association between the candidate gene haplotype and NPQ(T) were analyzed by the Kruskal–Wallis test, run in the R software.

## 3. Result

### 3.1. Genome Resequencing of 92 Rice Varieties

To characterize genomic variation in the rice accession collection, whole-genome resequencing data were generated for the 173 rice cultivars, which were collected from three rice subpopulation around the world (indica: 101, japonica: 58, intermediate: 14) (Supplementary Table S1). A total of 466.7 Gb high-quality sequence data (64.4 billion paired-end reads) were obtained, ranging from 0.88 to 16.52 Gb, with an average of 5.17 Gb for each line (Supplementary Table S1). The sequence reads for each line were aligned to the v4.1 draft reference genome of 'Nipponbare' (https://www.ebi.ac.uk/ena/browser/view/GCA_001433935.1 (accessed on 10 August 2022)). The coverage depth in each line ranged from 4.6- to 42.18-fold, with a mean of 13.59-fold (Supplementary Table S1). A total of 1,566,981 SNPs were identified based on alignment to the reference genome. (Figure 1; Supplementary Table S2). The Ts/Tv ratio (transitions/transversions) ranged from 2.00 to 2.29, with a mean of 2.23. The SNPs were distributed on all 12 chromosomes of rice with a density of 4.17/kb, ranging from 3.24/kb on chromosome 04 to 4.96/kb on chromosome 03 (Figure 1; Supplementary Table S2).

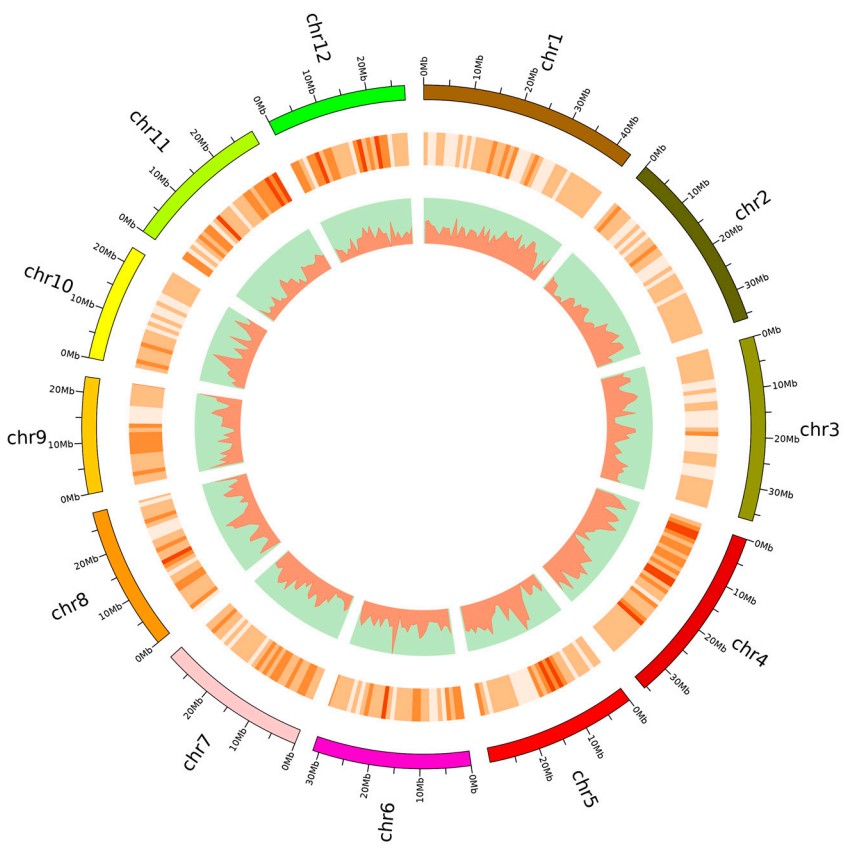

**Figure 1.** Display of single-nucleotide polymorphism (SNPs) identified in the population. The rings from outside to inside indicate the individual chromosomes, SNP distribution, and genomic GC distribution. SNPs were calculated by 1000 kb sliding window across each chromosome.

To gain insights into the potential effects of SNPs, all 1,566,981 detected SNPs were functionally annotated (Supplementary Table S3). Of these, most SNPs (52.04%) were located in intergenic regions, 16.49% were located within the 1 kb upstream/downstream regions of genes, 912 (0.06%) were located within ncRNA regions, and 492,298 (31.42%) were located within genic regions. Of those in genic regions, 270,249 (17.25%) were located in introns, which is more than in exons (133,368, 8.51%). Notably, 66,335 (4.23%) SNPs were nonsynonymous substitutions, 436 (0.03%) altered splicing, 845 (0.05%) induced gain of stop codon, and 170 (0.01%) induced loss of stop codon (Supplementary Table S3). These SNPs may have significant effects on gene function and could be used to identify the likely candidate genes hereafter.

### 3.2. Population Structure, Linkage Disequilibrium, and Phenotypic Variation

To understand the overall genetic relationship in this population, we explored the phylogenetic relationship and performed PCA of 173 accessions using randomly selected SNP markers. The neighbor-joining tree revealed that most of the accessions, belonging to the two subspecies of indica and japonica rice, were clustered with each other (Figure 2a). Additionally, 14 intermediate accessions, which may be from historical hybrids between indica and japonica rice, were randomly mixed with the two groups of divergent subspecies (Figure 2a). This result was also supported by the PCA, in which the PC1 explained 32.25% and PC2 explained 10.19% of the genetic variation within these 173 rice lines (Figure 2b). Based on the $r^2$ value, which declined to half of the maximum value, the linkage disequilibrium (LD) decay for 173 rice varieties was estimated at 52.5 kb (Figure 2c), indicating that the rice lines exhibited a moderate LD [16].

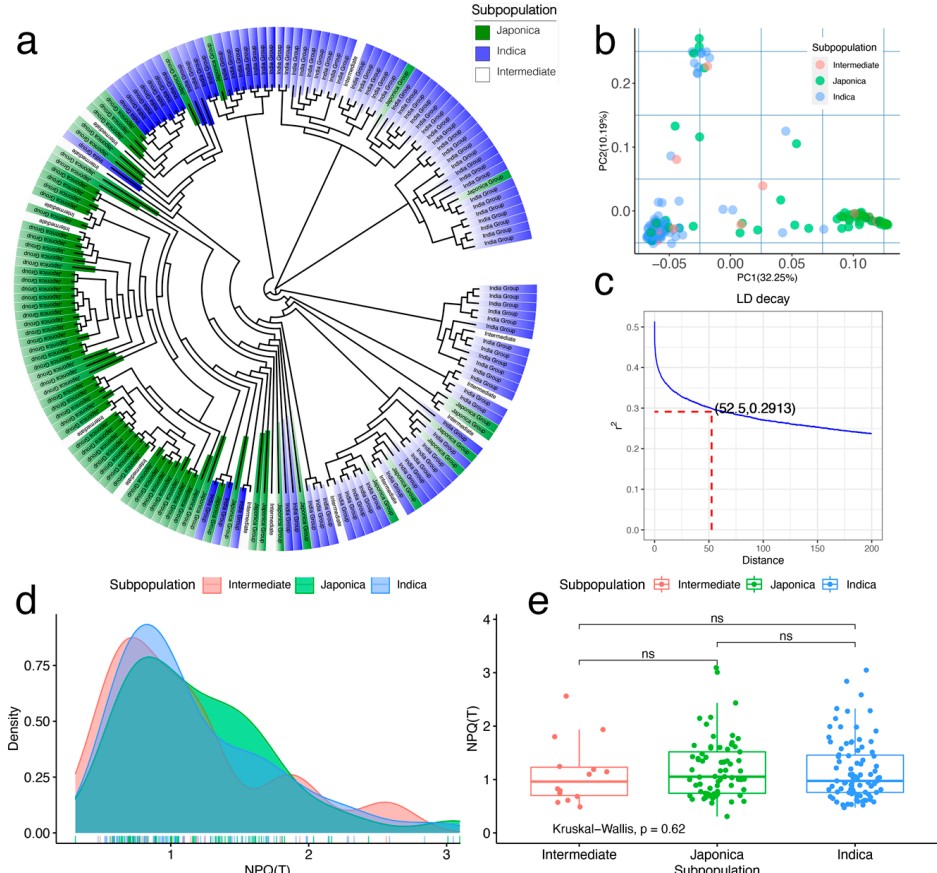

**Figure 2.** Features of the rice population of 173 rice varieties. (**a**) Neighbor-joining phylogenetic tree of the 173 accessions inferred from 112,408,956 SNPs. (**b**) PCA plots of the first two components of 173 accessions. (**c**) Genome-wide average linkage disequilibrium (LD) decay of 173 rice varieties. (**d**) Frequency distribution histogram of the NPQ(T) in flag leaf of three rice groups. (**e**) Comparison of the NPQ(T) among the three rice groups.

Furthermore, we investigated the phenotypic variation of the NPQ(T) of 173 rice varieties in 2019. The distribution and descriptive statistical analysis in different subpopulations and the entire population are presented in Figure 2d,e and Table 1. The NPQ(T) has a large variation of 10-fold between lines with the lowest (0.31) and highest (3.1) values in the whole population. Comparisons among the three groups revealed that there were no significant differences among the indica, japonica, and intermediate rice.

**Table 1.** Phenotypic variation of NPQ values in different groups.

| Subpopulation | No. of Line | Mean | SD | Median | Max. | Min. |
|---------------|-------------|------|------|--------|------|------|
| Indica | 90 | 1.16 | 0.56 | 0.97 | 3.05 | 0.47 |
| Intermediate | 14 | 1.12 | 0.55 | 1.05 | 3.1 | 0.31 |
| Japonica | 69 | 1.21 | 0.6 | 0.96 | 2.56 | 0.49 |
| All | 173 | 1.18 | 0.56 | 1.02 | 3.1 | 0.31 |

### 3.3. Genome-Wide Association Analysis (GWAS) for NPQ(T)

To detect the genetic basis of the NPQ(T), GWAS was performed using the EMMAX method, based on all 1,566,981 SNPs. Three signals, distributed on chromosomes 02, 05, and 07, were detected to be associated with the NPQ(T) by the significance threshold of $-\log10(P) = 6$ (Figure 3a). The Q-Q plots (quantile–quantile plots) indicate that the model fits the data fairly well (Figure 3b). We calculated the PVE of 100 kb before and

after the peak signal; afterwards, we focused on the associated signal on chromosome 02 (Supplementary Table S5). Here, we estimated a candidate region of 194.9 kb from 4498 kb to 4698 kb by using $r^2 > 0.8$ (Figure 3c).

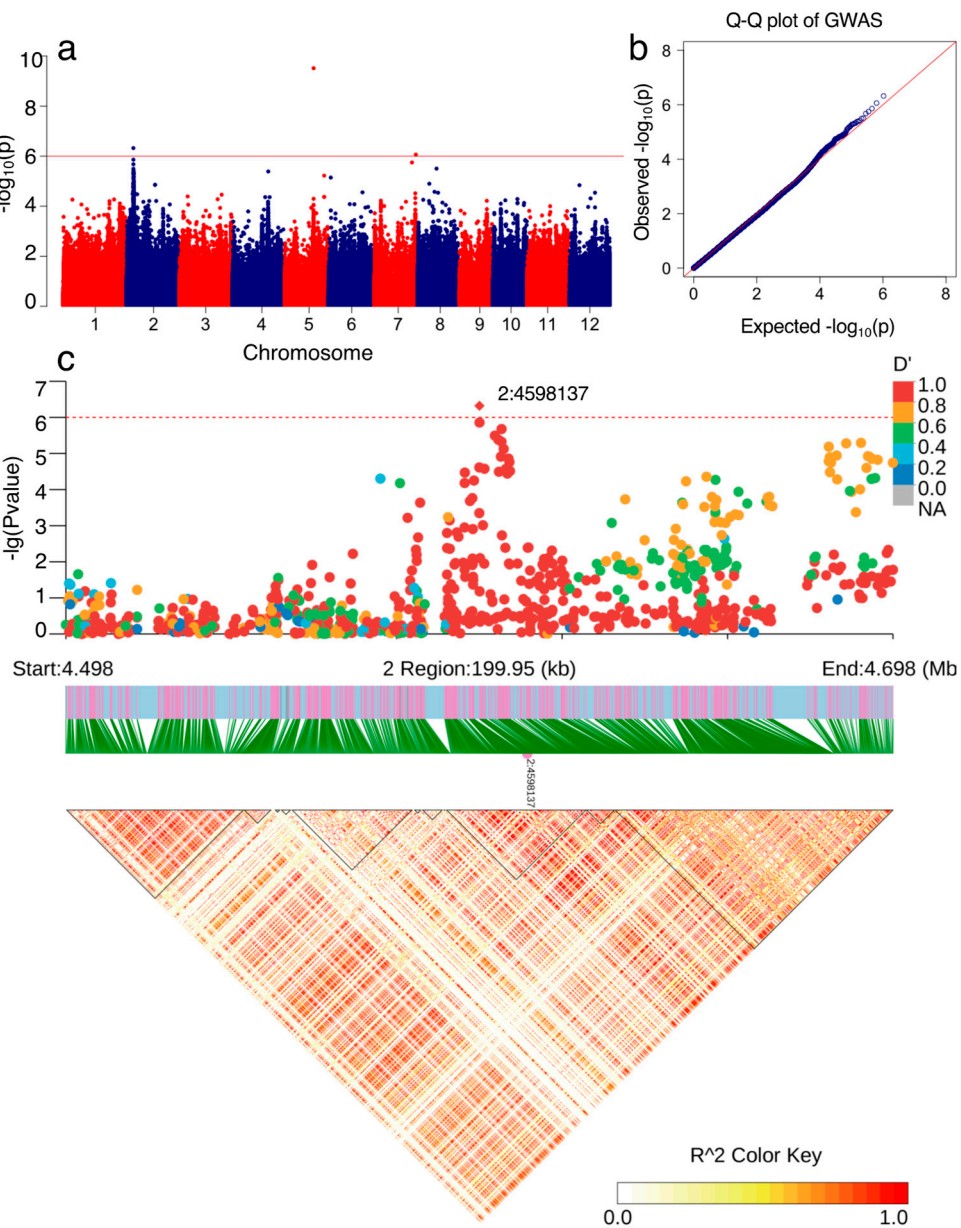

**Figure 3.** GWAS of NPQ(T) for the whole population in 2019. (**a**) Manhattan plot for NPQ(T). Dashed line represents the significance threshold ($-\log 10(P) = 6$). (**b**) Q−Q plot for NPQ(T) (**c**). Local Manhattan plot (top) and LD heatmap (bottom) surrounding the peak on chromosome 02.

### 3.4. Candidate Genes Underlying the Associated Signal on Chromosome 02

A total of 28 genes were found in the above-mentioned candidate region of 194.9 kb on chromosome 02, of which 16 genes have known annotations in the rice database (Supplementary Table S4). The associations between the functional variations of these target genes and NPQ(T) variation were investigated in the association panel. Of these, it was found that only SNPs in *Os02g0184100* were significantly associated with NPQ(T) variation. Six SNPs located in the promoter region of *Os02g0184100* formed three haplotypes (Figure 4a). The NPQ(T) value of inbred lines carrying haplotype 1 ($1.25 \pm 0.60$) was significantly higher than lines carrying haplotype 2 (($0.95 \pm 0.35$) ($p = 0.046$) (Figure 4b)). Five nonsynonymous

SNPs located in the exons of *Os02g0184100* formed three major haplotypes (Figure 4c). The NPQ(T) value of inbred lines carrying haplotype 1 (0.97 ± 0.37) was significantly lower than lines carrying haplotype 2 ((1.37 ± 0.63) (*p* = 0.00095) (Figure 4c)). Furthermore, we investigated the expression pattern of *Os02g0184100* in RiceXPro (https://ricexpro.dna.affrc.go.jp (accessed on 10 August 2022)). It was found that *Os02g0184100* was highly expressed in the flag leaves at 83 DAT (Figure 4d), which is close to the stage when we investigated the NPQ(T) in our study. Notably, the expression of *Os02g0184100* has a circadian variation in the flag leaves, with a high expression level at noon (Figure 4e). These results demonstrated that *Os02g0184100* is the most likely candidate gene for the association locus.

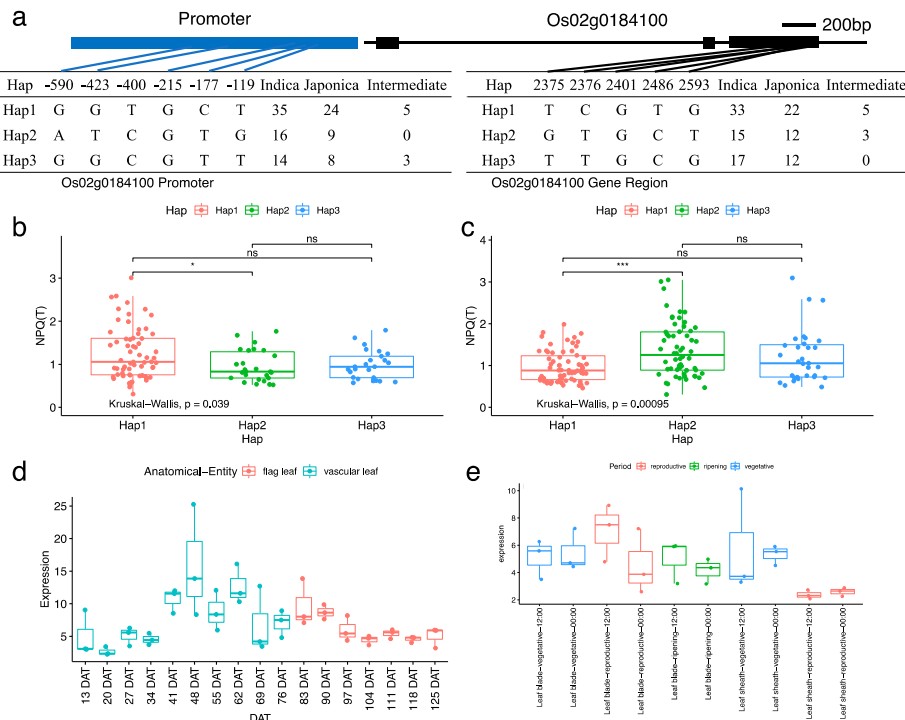

**Figure 4.** Candidate genes for NPQ(T) underlying the associated loci on chromosome 02. (**a**) Haplotype analysis of *Os02g0184100*. (**b**,**c**) Boxplots for NPQ(T) based on the genotypes of promoter (**b**) and gene region (**c**) of *Os02g0184100*. (**d**) Expression pattern of *Os02g0184100* in the vascular leaf and flag leaves of rice at different growth stages. (**e**) Diurnal expression pattern of *Os02g0184100* in the flag leaves.

## 4. Discussion

Photoprotection is a well-defined plant process which helps to prevent the deleterious effects of high and excess light on plant cells, especially within the chloroplast. Molecular components of chloroplast photoprotection are closely aligned with those of photosynthesis and they influence the productivity together [34]. Chlorophyll fluorescence analysis is one of the most powerful and widely used techniques to study the effect of stresses on the photosynthetic process. Previous studies have demonstrated that NPQ/NPQ(T) is an important photoprotective strategy for plants to adapt to the highly excessive natural illumination during the growing season as well as the dynamic light microenvironments inside the canopy [15,35]. The molecular mechanism underlying NPQ/NPQ(T) has been extensively investigated in Arabidopsis [11,19,36]. However, the NPQ(T) has yet to be studied comprehensively in the natural rice population. In this study, we investigated the NPQ(T) variation in a population consisting of diverse 173 rice accessions. The NPQ(T) displayed a 10-fold difference in our collected panel (Table 1), suggesting that NPQ(T) had an extensive phenotypic variation in the rice cultivars. No significant difference was

observed between indica and japonica subspecies (Figure 2e). This result reveals that the NPQ(T) was not selected during the domestication of rice.

In the present study, we obtained a total of 466.7 Gb high-quality sequence data and identified 1,566,981 SNPs based on alignment to the reference genome (Figure 1). The high marker density enabled us to perform a more exhaustive GWAS for the NPQ(T), which facilitated the identification of a more complete set of likely candidate genes responsible for NPQ(T). Three loci were identified to be associated with NPQ(T), distributed on chromosomes 02, 05, and 07 (Figure 2). Compared with the traditional QTL mapping approach, our GWAS provided much higher resolution, facilitating candidate gene identification.

We focused on the loci on chromosome 02 and detected a total of 28 genes for NPQ(T). Close to the GWAS peak signals, we found some candidate genes that might be involved in processes of electron transport, buildup of proton gradient, PSII light-harvesting antenna structure, and light-harvesting complex II (LHCII) re-arrangements that trigger and regulate the NPQ/NPQ(T) scenario (Supplementary Table S4) [36]. The association between haplotype and phenotype has been proved to be a useful way to identity the candidate genes underlying the associated signals in plants [37,38]. In our study, the most likely candidate genes were uncovered by testing the correlation between functional polymorphisms and phenotypic variation and expression pattern in the flag leaves. We found that the NPQ(T) was correlated with the haplotype of *Os02g0184100* (Figure 4). Furthermore, by querying the gene expression, we found that *Os02g0184100* had a high expression level and a certain circadian rhythm in flag leaves. *Os02g0184100* encodes the phosphopantothenoylcysteine decarboxylase subunit VHS3. The phosphopantothenoylcysteine decarboxylase participates in the synthesis of CoA [39]. Acetyl-CoA is an important intermediate metabolite of energy metabolism, such as the Calvin cycle of photosynthesis [40]. Phosphopantothenoylcysteine has been reported to have salt resistance and antioxidant effects in apple [41]. NPQ is also proved to be an antioxidant photoprotective process [34], so we suggested that *Os02g0184100* may affect the photoprotection process of photosynthesis. This is consistent with the fact that the *Os02g0184100* expression in the daytime is higher than that in the nighttime. These results suggest that *Os02g0184100* might be the likely candidate gene associated with NPQ(T). NPQ/NPQ(T) is a consequence of complicated biological processes and its mechanism remains unclear; more detailed experimental analyses are needed to confirm the function of candidate genes in photoprotection [42,43]. However, the significant loci and likely candidate genes uncovered in the current study now provide us a comprehensive understanding of the genetic architecture of NPQ(T), and this in turn provides stronger evidence for the genetic improvement of rice photoprotection.

**Supplementary Materials:** The following are available online at https://www.mdpi.com/article/10.3390/agronomy12123216/s1, Figure S1: Changes in daily temperature and precipitation during the whole growth period of rice under natural conditions. Table S1: Summary of 173 varieties and resequencing data. Table S2: Distribution of SNPs on 12 chromosomes of Oryza sativa. Table S3: The annotation of SNPs in Oryza sativa. Table S4: The candidate genes for the associated loci detected by GWAS. Table S5: PVE of SNP site (100 kb before and after the signals).

**Author Contributions:** Conceptualization, Y.W. and J.H.; methodology, S.L.; software, Z.X.; validation, S.L., Z.X. and F.W.; formal analysis, F.W.; investigation, Y.W. and S.L.; resources, D.X. and F.W.; data curation, Y.W.; writing—original draft preparation, Y.W. and S.L.; writing—review and editing, J.H.; visualization, Z.Z.; supervision, S.L.; project administration, J.H.; funding acquisition, J.H. All authors have read and agreed to the published version of the manuscript.

**Funding:** This work was supported by Independent project of the National Key Research and Development Program of China (No. 2022YFD2300700) and Chinese National Natural Science Foundation (No. 31671620). The sponsors had no role in the design or conception of the study; the collection, management and analysis of the data; the preparation, writing and review of the manuscript; or the decision to submit the manuscript for publication.

**Data Availability Statement:** All data in this article are available. The DNA sequences raw data of samples were deposited in the National Center for Biotechnology Information (NCBI), accession:

PRJNA894819. Other phenotypic and gene expression data are included in this article and its Supplementary Materials.

**Acknowledgments:** We are thankful to the Key Laboratory of Crop Ecophysiology and Farming System in the Middle Reaches of the Yangtze River, Ministry of Agriculture, College of Plant Science and Technology, Huazhong Agricultural University.

**Conflicts of Interest:** The authors declare no conflict of interest. The funders had no role in the design of the study; in the collection, analyses, or interpretation of data; in the writing of the manuscript, or in the decision to publish the results.

## Abbreviations

| | |
|---|---|
| GWAS | Genome-wide association study |
| LD | Linkage disequilibrium |
| NPQ(T)/NPQ | Non-photochemical quenching |
| SNP | Single-nucleotide polymorphism |

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
