# Peer review of "Genome-Wide Association Study for Non-Photochemical Quenching Traits in Oryza sativa L."

_agronomy, doi:10.3390/agronomy12123216_

Round 1
Reviewer 1 Report
While the concept of the paper is of general interest, following the introduction section the quality of the manuscript is low.
In particular the materials and methods lack appropriate details and in many instances results are presented that have not been described. In it's current description it would not be possible to reproduce the authors results following the description of their methodology.
There are several points in the manuscript where information is listed without context (see lines 87-89; 130-134), or is difficult to interpret. Information pertaining to the materials and methods are listed in the introduction and results inappropriately, which makes the manuscript difficult to follow (See lines 61-64; 140-153).
Tests presented in figures are not described in the methodology (Example the Kruskal-Wallis test in Figure 3. B). Structural equation modelling is mentioned in the methodology but results are not presented.
The population genetics analysis exhibits several errors/misinterpretations or a lack of understanding of the underlying concepts of the methodology employed. These are exploratory techniques to define the number of clusters and the authors do not provide sufficient support for their interpretation or evidence that the analysis was conducted correctly. The groups appear to be based upon pre-defined physiological groups and not genetic information.
The neighbor-joining tree lacks bootstrapping support and it is unclear what the confidence or optimal tree grouping is. It is unclear if the tree is a summary of all replications or merely random sample. No support values are supplied.
The STRUCTURE analysis does not appear to have been replicated or described in full. The authors fail to provide justification for a limited K range when the neighbor joining tree shows multiple potential sub-divisions. No burn-in or number of Marchov chain Monte Carlo iterations are listed. Confidence in groupings is unknown as common interpretive statistics such as ln-deltaK/Evanno’s method are not presented.
The principal component analysis does not indicate how many PC's are relevant, no interpretive information such as a scree plot is presented. The authors mention they calculated 3 components but provide no justification, nor do they indicate how much variation the third component explains. Furthermore, the clusters presented in the PCA of the first two components do not coincide with the population clusters the authors define which directly contradicts their results.
Including this information in the manuscript feels tagged on, and as it is unclear if it had been performed correctly does not contribute to the narrative the authors are presenting.
The authors do not appear familiar with the terminology LOD scores which is found in abundance in the GWAS literature and mistakenly refer to them as P-values. LOD scores are an interpretation of P-values. Significance thresholds are intended to limit the rate of false-positives but the authors present hundreds of significant SNPs suggesting a high degree of false-positive results. The authors should review association mapping statistical theory.
It is unclear is comparing the data they generated to downloaded sequences are of comparable formats and if the data can be analyzed jointly without significant bias.
Figures such as the Manhattan plot lack appropriate axes to interpret results.
I have not commented further on grammar and formatting issues due to time constraints.
Author Response
Response to Reviewer Comments
Dear reviewer,
Thank you for your valuable and constructive comments on our article. Based on your suggestions, we have made major revision the information and results. The relevant background information of NPQ (T) was supplemented and the candidate genes were further analyzed and confirmed. We have supplemented materials, methods and specific parameters related to population genetics and statistics. And we rewrote the discussion section and summarized and discussed the contents of the revised article. We hope that the revised manuscript is improved and reads better, and hopefully merits publication in Agronomy.
Thanks again and best regards.
The following is our reply to the specific modification suggestions:
Point 1: In particular the materials and methods lack appropriate details and in many instances results are presented that have not been described. In it's current description it would not be possible to reproduce the authors results following the description of their methodology.
Response 1: We have supplemented materials, methods and specific parameters related to population genetics and statistics, including NPQ (T) measuring instruments and protocols (see lines 86-89), population genetic analysis software, versions and methods (see lines 110-123), and statistical testing methods (see lines 132-135).
Point 2: There are several points in the manuscript where information is listed without context (see lines 87-89; 130-134), or is difficult to interpret. Information pertaining to the materials and methods are listed in the introduction and results inappropriately, which makes the manuscript difficult to follow (See lines 61-64; 140-153).
Response 2: We have rewritten the summary and supplemented the parts with incoherent logic and modified or deleted the wrong parts.
Point 3: Tests presented in figures are not described in the methodology (Example the Kruskal-Wallis test in Figure 3. B). Structural equation modelling is mentioned in the methodology but results are not presented.
Response 3: Solved, and we deleted the structural equation modeling part.
Point 4: The population genetics analysis exhibits several errors/misinterpretations or a lack of understanding of the underlying concepts of the methodology employed. These are exploratory techniques to define the number of clusters and the authors do not provide sufficient support for their interpretation or evidence that the analysis was conducted correctly. The groups appear to be based upon pre-defined physiological groups and not genetic information.
The neighbor-joining tree lacks bootstrapping support and it is unclear what the confidence or optimal tree grouping is. It is unclear if the tree is a summary of all replications or merely random sample. No support values are supplied.
The STRUCTURE analysis does not appear to have been replicated or described in full. The authors fail to provide justification for a limited K range when the neighbor joining tree shows multiple potential sub-divisions. No burn-in or number of Marchov chain Monte Carlo iterations are listed. Confidence in groupings is unknown as common interpretive statistics such as ln-deltaK/Evanno’s method are not presented.
The principal component analysis does not indicate how many PC's are relevant, no interpretive information such as a scree plot is presented. The authors mention they calculated 3 components but provide no justification, nor do they indicate how much variation the third component explains. Furthermore, the clusters presented in the PCA of the first two components do not coincide with the population clusters the authors define which directly contradicts their results.
Including this information in the manuscript feels tagged on, and as it is unclear if it had been performed correctly does not contribute to the narrative the authors are presenting.
Response 4: We have re-analyzed and mapped the results of population genetic analysis (see Figure 2, lines 170-182), analyzed and compared them according to the sub populations of indica rice and japonica rice, and deleted STRUCTURE analysis results. Relevant parameters of PCA, neighbor-joining tree and LD decay have supplemented in the paper (see lines 113-117).
Point 5: The authors do not appear familiar with the terminology LOD scores which is found in abundance in the GWAS literature and mistakenly refer to them as P-values. LOD scores are an interpretation of P-values. Significance thresholds are intended to limit the rate of false-positives but the authors present hundreds of significant SNPs suggesting a high degree of false-positive results. The authors should review association mapping statistical theory.
Response 5: We have corrected the description problem related to population genetics.
Point 6: It is unclear is comparing the data they generated to downloaded sequences are of comparable formats and if the data can be analyzed jointly without significant bias.
Response 6: We have provided relevant information of all varieties in the supplementary Table S1, and use mixed linear model for GWAS analysis to avoid bias.
Point 7: Figures such as the Manhattan plot lack appropriate axes to interpret results.
Response 7: We have revised the figure3 correlation diagram and description.

Reviewer 2 Report
The manuscript entitled “Genome-wide association study for non-photochemical quenching traits in Oryza sativa L.” by Youbo et al. aimed to identify putative candidate genes associated with NPQ (T) through genome wide association study. The authors have used multiple methods to conduct GWAS. They have provided a list of candidate genes related to NPQ (T) traits in rice flag leaves. The authors suggested that the significant differences in NPQ (T) of rice varieties could provide theoretical basis and technical guidance for the selection and breeding of high-photosynthesis-efficiency varieties. The research work is important, as it is associated with the enhancement of photosynthetic ability of rice leaves that leads to higher yield.
However, I have the following major comments for the authors:
1. The present study mainly focused on the NPQ (T) of the flag leaf of rice. Hence, the authors need to mention specifically flag leaf instead of rice leaves. In the abstract the authors have mentioned that “Furthermore, 4 putative candidate genes analyzed: Os02g0182100(RR24) 19 and Os02g0182800(HOS58) were highly expressed in rice leaves, Os02g0181900(CLPB-M) and 20 Os02g0181300(WRKY71) were lowly expressed in rice leaves.” Please change it to flag leaf in rice. Please also change rice leaves to flag leaf of rice wherever appropriate throughout the manuscript. The readers will be confused, as you have shown in the heat map for matured leaf and young leaf and you have mainly focused on flag leaf.
2. Introduction needs major revision, as there is not much information on the subject.
3. The authors have used multiple methods to conduct GWAS and identified 29 significant associations. However, there is insufficient information on the candidate genes that they have identified. To verify the association between the candidate gens and NPQ (T) traits in rice flag leaf, the authors need to do the full length sequencing analysis of those candidate genes. They need to provide a detailed information and sequence analysis of those candidate genes.
4. Is there any difference in the phenotype of the seedlings? Please provide a photo of those seedlings if you have any.
5. Discussion needs major revision.
6. Please discuss the suggested mechanism and function of those candidate genes in enhancing NPQ (T) in flag leaf of rice. If possible, validate your analysis results to confirm the roles played by those candidate genes. Are those genes involved in any specific metabolic processes, phytohormone-signaling pathways etc.? If so, please discuss.
7. Please provide a summary figure describing the function of the candidate genes.
8. Please refine the language throughout the manuscript.
For example in the abstract, the authors have mentioned that: “The different the groups significant differences in NPQ (T) of rice varieties can provide theoretical basis and technical guidance for the selection and breeding of high-photosynthesis-efficiency varieties.”….please rephrase it.
Author Response
Response to Reviewer Comments
Dear reviewer,
Thank you for your valuable and constructive comments on our article. Based on your suggestions, we have made major revision the information and results. The relevant background information of NPQ (T) was supplemented and the candidate genes were further analyzed and confirmed. We have supplemented materials, methods and specific parameters related to population genetics and statistics. And we rewrote the discussion section and summarized and discussed the contents of the revised article. We hope that the revised manuscript is improved and reads better, and hopefully merits publication in Agronomy.
Thanks again and best regards.
The following is our reply to the specific modification suggestions:
Point 1: The present study mainly focused on the NPQ (T) of the flag leaf of rice. Hence, the authors need to mention specifically flag leaf instead of rice leaves. In the abstract the authors have mentioned that “Furthermore, 4 putative candidate genes analyzed: Os02g0182100(RR24) 19 and Os02g0182800(HOS58) were highly expressed in rice leaves, Os02g0181900(CLPB-M) and 20 Os02g0181300(WRKY71) were lowly expressed in rice leaves.” Please change it to flag leaf in rice. Please also change rice leaves to flag leaf of rice wherever appropriate throughout the manuscript. The readers will be confused, as you have shown in the heat map for matured leaf and young leaf and you have mainly focused on flag leaf.
Response 1: We carefully checked and revised the content of the full text, so that “flag leaf” description can be unified.
Point 2: Introduction needs major revision, as there is not much information on the subject.
Response 2: We have made major revision the information, introduced the relationship between NPQ (T) and photosynthesis (see lines 81-97).
Point 3: The authors have used multiple methods to conduct GWAS and identified 29 significant associations. However, there is insufficient information on the candidate genes that they have identified. To verify the association between the candidate gens and NPQ (T) traits in rice flag leaf, the authors need to do the full length sequencing analysis of those candidate genes. They need to provide a detailed information and sequence analysis of those candidate genes.
Response 3: We agree with the claims given by the reviewer, full length sequencing analysis is a powerful candidate gene verification method, but we can identify the variation according to the second generation sequencing, and also verify the candidate genes through the correlation between haplotype and phenotype.
Point 4: Is there any difference in the phenotype of the seedlings? Please provide a photo of those seedlings if you have any.
Response 4: Due to the large number of varieties in the population and the inconsistency of leaf age among varieties. In order to unify the standard, we selected the flag leaf for investigation and research, and did not investigate the growth of young leaves.
Point 5: Discussion needs major revision.
Response 5: We have rewritten the discussion (see lines 286-345).
Point 6: Please discuss the suggested mechanism and function of those candidate genes in enhancing NPQ (T) in flag leaf of rice. If possible, validate your analysis results to confirm the roles played by those candidate genes. Are those genes involved in any specific metabolic processes, phytohormone-signaling pathways etc.? If so, please discuss.
Response 6: In the result section (see lines 253-269), we have carried out gene annotation, haplotype analysis and expression profile query for 28 important associated genes. The gene Os02g0184100 was finally determined and its possible function was discussed (see lines 332-339).
Point 7: Please provide a summary figure describing the function of the candidate genes.
Response 7: We have described the function of candidate genes in the supplementary Table S4.
Point 8: Please refine the language throughout the manuscript.
For example in the abstract, the authors have mentioned that: “The different the groups significant differences in NPQ (T) of rice varieties can provide theoretical basis and technical guidance for the selection and breeding of high-photosynthesis-efficiency varieties.”….please rephrase it.
Response 8: In the process of revising and rewriting the article, we corrected the errors in the language part.

Round 2
Reviewer 2 Report
Thank you authors for the revision.
However, I have the following comments for the authors:
1. In the original manuscript the authors have mentioned in the abstract that “A genome-wide association study (GWAS) utilizing 1566 981 high-quality SNPs and 532 401 Indels, Identified 29 significant associations. Furthermore, 4 putative candidate genes analyzed: Os02g0182100 (RR24) and Os02g0182800 (HOS58) were highly expressed in rice leaves, Os02g0181900 (CLPB-M) and Os02g0181300 (WRKY71) were lowly expressed in rice leaves. However, in the revised manuscript the authors have mentioned that” Furthermore, one likely candidate gene Os02g0184100, underlying the associated signal on chromosome 02, were uncovered by identifying both the expression pattern in flag leaves and testing the correlation between functional polymorphisms and phenotypic variation.” Now they have removed three genes. Could you please explain the reason for it?
2. In the revised manuscript, the authors have mentioned in the discussion line 246-247 that “Three loci were identified to be associated with NPQ (T), distributed on the chromosome 02, 05 and 07, (Figure 2).” In the line 250, they have mentioned that, “We focused on the loci on chromosome 02 and detected a total of 28 genes for NPQ (T).” Could you please explain why the authors have decided to focus only on chromosome 02, but not the genes identified on chromosome 05 and 07? You also do not have strong explanation regarding the candidate gene identified on chromosome 02.
3. In the results, the authors have mentioned in line 214 that “Notably, the expression of Os02g0184100 has a circadian variation in the flag leaves, with high expression level at noon (Figure 4f).” Figure 4f or 4e…please check.
4. Discussion still needs improvement. In the discussion, the explanation for the probable role of the single candidate gene is very limited. You need more explanation.
Author Response
Response to Reviewer Comments
Dear reviewer,
Thank you for your valuable and constructive comments on our article. Based on your suggestions, we have supplemented relevant content from the aspects of the Calvin cycle and antioxidants. Supplemented the corresponding PVE of SNP, and made corresponding supplements in the manuscript. We hope that the revised manuscript is improved and reads better, and hopefully merits publication in Agronomy.
Thanks again and best regards.
The following is our reply to the specific modification suggestions:
Point 1: In the original manuscript the authors have mentioned in the abstract that “A genome-wide association study (GWAS) utilizing 1566 981 high-quality SNPs and 532 401 Indels, Identified 29 significant associations. Furthermore, 4 putative candidate genes analyzed: Os02g0182100 (RR24) and Os02g0182800 (HOS58) were highly expressed in rice leaves, Os02g0181900 (CLPB-M) and Os02g0181300 (WRKY71) were lowly expressed in rice leaves. However, in the revised manuscript the authors have mentioned that” Furthermore, one likely candidate gene Os02g0184100, underlying the associated signal on chromosome 02, were uncovered by identifying both the expression pattern in flag leaves and testing the correlation between functional polymorphisms and phenotypic variation.” Now they have removed three genes. Could you please explain the reason for it?
Response 1: Since the previous database is mainly aimed at Arabidopsis, the analysis of rice expression patterns is inaccurate. In the revised, we used the new database RiceXPro. And systematically analyzed haplotypes and expression patterns of all genes, and finally selected this gene.
Point 2: In the revised manuscript, the authors have mentioned in the discussion line 246-247 that “Three loci were identified to be associated with NPQ (T), distributed on the chromosome 02, 05 and 07, (Figure 2).” In the line 250, they have mentioned that, “We focused on the loci on chromosome 02 and detected a total of 28 genes for NPQ (T).” Could you please explain why the authors have decided to focus only on chromosome 02, but not the genes identified on chromosome 05 and 07? You also do not have strong explanation regarding the candidate gene identified on chromosome 02.
Response 2: First, we can clearly see from the Manhattan map that the gene linkage of chromosomes 05 and 07 is not as good as that of chromosome 02. Second, we compared the PVE value (Table S5) of the SNP. So we thought they may be genotyping errors or mutations. Finally, the most likely chromosome 02 was selected and explained in the manuscript (see lines 123-124, 193-195).
Point 3: In the results, the authors have mentioned in line 214 that “Notably, the expression of Os02g0184100 has a circadian variation in the flag leaves, with high expression level at noon (Figure 4f).” Figure 4f or 4e…please check.
Response 3: Done
Point 4: Discussion still needs improvement. In the discussion, the explanation for the probable role of the single candidate gene is very limited. You need more explanation.
Response 4: We have supplemented relevant content from the aspects of the Calvin cycle and antioxidants (see lines 264-272).
